# Maternal Environmental Light Conditions Affect the Morphological Allometry and Dispersal Potential of *Acer palmatum* Samaras

**Bin J. W. Chen** [1,*], **Xinyu Wang** [1], **Yuting Dong** [2], **Heinjo J. During** [3], **Xia Xu** [1] and **Niels P. R. Anten** [4]

[1] College of Biology and the Environment, Nanjing Forestry University, Nanjing 210037, China; Wangxinyu1007@outlook.com (X.W.); xuxia.1982@yahoo.com (X.X.)
[2] Department of Terrestrial Ecology, Netherlands Institute of Ecology, 6708PB Wageningen, The Netherlands; y.dong@nioo.know.nl
[3] Institute of Environmental Biology, Utrecht University, 3508TB Utrecht, The Netherlands; H.J.During@uu.nl
[4] Crop & Weed Ecology Group, Centre for Crop Systems Analysis, Wageningen University, 6700AK Wageningen, The Netherlands; niels.anten@wur.nl
* Correspondence: bin.chen@outlook.com

**Abstract:** Seed dispersal plays critical roles in determining species survival and community structures. Since the dispersal is biologically under maternal control, it is hypothesized that intraspecific variation of dispersal potential and associated traits of seeds (diaspores) should be influenced by maternal habitat quality. We tested this hypothesis by examining the effects of maternal environmental light condition on morphological traits and descending performance of nearly 1800 wind-dispersed samaras collected from maple species *Acer palmatum*. Results showed that samaras produced by trees from shaded microhabitats had greater dispersal potential, in terms of slower terminal velocity of descent, than those produced in open microhabitats. This advantage was largely attributed to morphological plasticity. On average, samaras produced in shaded microhabitats, as compared to those produced in open habitats, had lower wing loading by only reducing weight but not area. In allometric details, in the large size range, samaras from shaded microhabitats had larger areas than those from open microhabitats; in the small size range, samaras from shaded microhabitats had wider wings. These findings suggest that greater dispersal potential of samaras in response to stressful maternal light environment reflected an active maternal control through the morphological allometry of samaras.

**Keywords:** autorotation; diaspore; habitat selection theory; informed dispersal theory; Japanese maple; maternal environmental effect; seed dispersal; terminal velocity; wind dispersal

## 1. Introduction

Dispersal, a phenomenon that depicts an individual unidirectionally moving away from its mother or place of birth, is ubiquitous in all organisms [1], and has long been the interests of ecologists [2,3]. For plants as sessile organisms, the dispersal of seeds is particularly important since it represents the predominant, probably also the only mobile stage of most terrestrial species throughout their lifespan [4,5]. From an evolutionary perspective, seed dispersal enhances the fitness of plants by avoiding disease- or predation-induced high offspring mortality near the parents, reducing offspring competition, reducing inbreeding, increasing chances of finding suitable habitats, and colonizing disturbed areas [6,7]. Therefore, seed dispersal plays fundamental roles in determining spatial distribution, gene flow, population growth, and thus the survival of a species, as well as maintaining species coexistence and diversity, and thus the community structure and dynamics [8–10]. Recently, seed dispersal is receiving increasing attention as it is also critical to the success of plants to cope with climate change and habitat fragmentation, and to the success of invasion [11–13].

Ample evidence shows a large variation in seed dispersal within species [14]. Although such a variation is believed to have important ecological and evolutionary consequences, our understanding of its drivers is still limited [5,15]. While much of this intraspecific variation in seed dispersal traits is believed to be due to intrinsic genetic factors, some studies also hypothesized that the growth environments of parental plants, maternal plants in particular, have a great contribution [16,17], the phenomenon of which is often coined as a maternal environmental effect (reviewed by Donohue [18]). From a biological perspective, this can be attributed to the fact that seed provision is a maternally-controlled process, and the embryo in a seed is fully surrounded by maternal tissues [19]. From an evolutionary perspective, seeds should be dispersed to suitable sites for better growth and reproduction in order to increase the fitness of both offspring and maternal plants. Both habitat selection theory [17] and informed dispersal theory [20] predict that since the quality of remote habitat is uncertain, seeds should stay nearby when the home site is suitable, but should be dispersed further away for a better chance of finding suitable sites when the home site is stressful [21]. However, to date, direct evidence of maternal environmental effects, especially from fields, on seed dispersal potential is still limited [10,12].

Due to the sessile lifestyle of plants, seeds are often passively dispersed by vectors, including wind, water and animals [3,22]. Among them, windborne ones represent the most common dispersal mode that exists in almost all terrestrial ecosystems [23], and often overwhelms (e.g., up to 70% of flora) in temperate communities [24]. As an adaptation, these windborne seeds are generally dispersed in the form of diaspore, i.e., a dispersal unit of a seed/fruit attached with some maternally-originated attributes [6,25].

Among various morphs of diaspores, "samara" as an achene attached with a wing-shaped attribute developed from an extension of ovary walls [26] is typically found in, e.g., elm, pine, maple and ash trees [27]. During descent, a symmetric samara (center of gravity locates in front of the aerodynamic center) glides without rotation, while an asymmetric samara (center of gravity locates near the terminal end of wing) demonstrates a unique autorotation motion [28]. In detail, it initially falls with an acceleration under gravity, but soon, a leading edge vortex generated by its autorotation motion produces a lifting force against the gravity force and decreases the descending rate until reaching an equilibrium state of stable descending rate, namely terminal velocity [29,30]. This terminal velocity is a key factor that determines dispersal potential, since dispersal distance of a diaspore (hereafter simply, "seed") by wind critically depends on the period of exposure to the horizontal wind current, and the duration of this period depends on how fast the seed descends [26]. Therefore, it evolutionarily represents a functional consequence of morphological adaptation in response to wind dispersal [6].

To date, most studies of asymmetric samaras (hereafter simply, samaras) have focused on the impacts of their geometries on the kinematic and aerodynamic characteristics of their autorotation motions and the consequences on dispersal efficiency [28,29,31–34]. However, understanding of the dispersal ecology is still limited [12]. For instance, it is largely unknown whether the dispersal potential of samaras can be affected by maternal environmental conditions, and to what extent these effects can be attributed to the morphological plasticity of samaras. Knowledge gained so far is generally no more than that the square root term of wing loading is linearly correlated with the terminal velocity of samaras [35,36], and trees which grew in different environments tend to produce samaras with different wing loading [25,37].

In the current study, we did a field investigation on the effects of maternal environmental light conditions on terminal velocity and associated morphological traits of samaras of a maple species *Acer palmatum* commonly planted in urban areas as ornamental trees. Since a heavily shaded growth condition not only directly reflects a remarkably lower level of aboveground light resources, but also may imply a high neighborhood density with intensive competitions for belowground soil resources, we expect that *A. palmatum* trees growing in a heavily shaded environment should experience much lower levels of

resource availability than trees growing in an unshaded environment. According to the habitat selection theory [17] and informed dispersal theory [20], we hypothesize that trees of *A. palmatum* which grew in shaded microhabitats should produce samaras with a suit of morphological traits that promote dispersal potential (in terms of a reduced terminal velocity) as compared to those which grew in open (i.e., unshaded) microhabitats. In addition, we are aware that since the morphological plastic responses of seeds to maternal environment can be caused either by an allometric variation (i.e., an environment-induced variation in size leads to a variation of morphological trait values, but the trait-size relationship still obeys the same allometric trajectory), or by a nonallometric variation (i.e., the trait-size relationships follow different allometric trajectories) [38]. Therefore, we further test whether the improved dispersal potential of *A. palmatum* samaras from shaded microhabitats is caused by a change of their sizes (i.e., allometric variation), or by a change of their morphological designs as an active maternal response (i.e., nonallometric variation).

## 2. Materials and Methods

### 2.1. Plant Materials

*Acer palmatum* (Aceraceae), also called "Japanese maple", is a deciduous subcanopy tree species in temperate forests native to China, Japan and Korea [39]. It has a long history of cultivation, and nowadays is still a popular small ornamental tree species widely used in the gardening and greening of parks, communities and green belts of roads in China. It has a phenology of flowering in April and May, and fruiting in September. As a typical maple species, its seeds are packaged in the form of samaras and are mainly dispersed by wind [40]. A fruit of *A. palmatum* consists of two samaras, which are mutually connected at their seed end and with a pedicel attached to the connection part. The dispersal initiates when matured samaras separate from each other and are detached from the pedicel.

### 2.2. Field Collection

The collection occurred in November when all samaras were mature in light brown-yellow colors, but still attached on the maternal trees. Three sampling sites, i.e., Gulou Campus of Nanjing University (118°46′26.05″ E, 32°03′25.53″ N), Xuanwu Lake Scenic Area (118°47′38.38″ E, 32°04′28.11″ N) and Zhongyang Road (affiliated green belts) (118°46′43.81″ E, 32°04′36.51″ N) representing different urban land use types, were selected in the downtown area of Nanjing, eastern China (Figure 1). In each site, ten trees of *A. palmatum* were selected (see basic information of the selected trees in Table 1, and their locations in Figure 1). Half (five trees) of them grew in typical shaded microhabitats, and the other half grew in typical open microhabitats. Here, the 'shaded microhabitat' was defined as an area being heavily and fully shaded by a large, intact and dense top-canopy layer consisting of crowns of several large canopy trees (typically from *Platanus acerifolia* and *Cinnamomum camphora*), so that focal trees were fully shaded and cannot receive direct sunlight. The 'open microhabitat' was defined as an area without any tall trees or buildings nearby, so that focal trees were not shaded and received full un-intercepted daylight. The selection of trees within a site was strictly controlled in terms of tree size (Table 1), variety and growth environment (except for light conditions) to minimize confounding factors. This, however, did not exclude variation in age between trees of similar size. As suggested by Hurlbert [41], the selected trees in shaded and open microhabitats were interspersed within each sampling site. It should also be noted that all selected *A. palmatum* trees were intentionally planted as ornamental trees rather than spontaneous. They regularly received gardening cares, though the frequencies and intensities of these cares were different among sites.

For each selected tree, the aim was to collect at least 50 undamaged, mature samaras. However, this was not always achieved, especially for trees which grew in shaded microhabitats (with the smallest sample size of 29). In total, 1789 samaras from 30 trees were collected and used in the analyses. These samaras were air-dried indoors for three weeks and stored in paper bags at room temperature before the measurements.

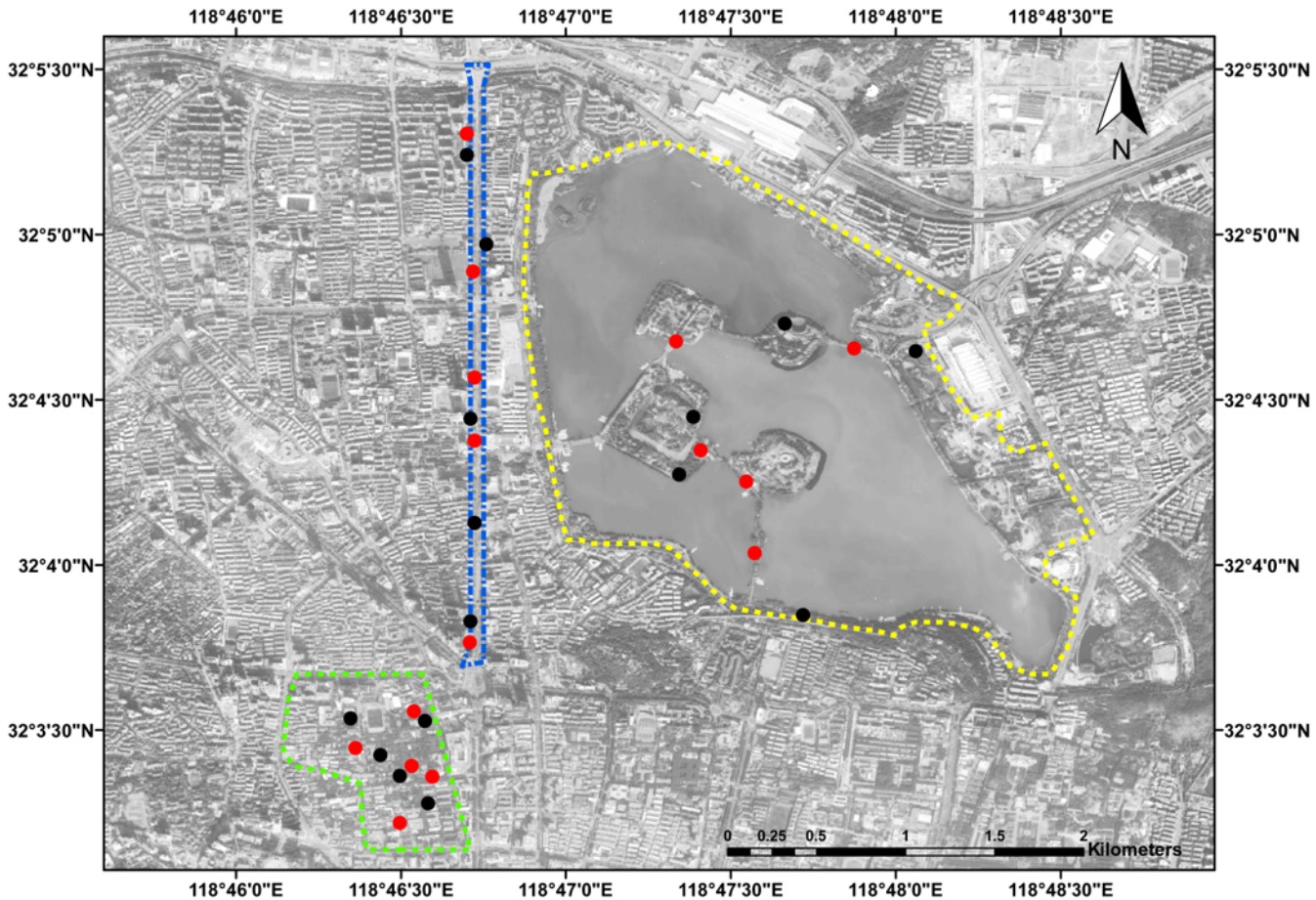

**Figure 1.** The locations of three sampling sites and 30 selected *Acer palmatum* trees in the downtown area of Nanjing, eastern China. Dash-circled areas in green, yellow and blue respectively represent Gulou Campus of Nanjing University, Xuanwu Lake Scenic Area and Zhongyang Road. Points in red and black respectively denote selected trees in open and shaded microhabitats.

**Table 1.** Basic information, i.e., mean $\pm$ 1 SD of the basal diameter (BD), height and crown diameter (CD) of selected *Acer palmatum* trees in three sampling sites.

| Sampling Site | Microhabitat | BD (cm) | Height (m) | CD (m) |
|---|---|---|---|---|
| NJU [1] | Open | 8.5 $\pm$ 0.4 | 3.0 $\pm$ 0.4 | 3.6 $\pm$ 0.3 |
| | Shaded | 8.4 $\pm$ 0.4 | 2.9 $\pm$ 0.2 | 3.2 $\pm$ 0.3 |
| XWL [2] | Open | 12.6 $\pm$ 1.7 | 3.4 $\pm$ 0.2 | 3.9 $\pm$ 1.0 |
| | Shaded | 12.4 $\pm$ 1.3 | 3.2 $\pm$ 0.3 | 3.4 $\pm$ 0.7 |
| ZYR [3] | Open | 6.6 $\pm$ 0.7 | 2.7 $\pm$ 0.6 | 2.9 $\pm$ 0.4 |
| | Shaded | 6.5 $\pm$ 0.9 | 2.6 $\pm$ 0.5 | 2.4 $\pm$ 0.2 |

[1] Gulou campus of Nanjing University. [2] Xuanwu Lake Scenic Area. [3] Zhongyang Road.

*2.3. Measurements*

All samaras were digitally imaged using a scanner (V330, Epson, Japan) with a resolution of 300 dpi. Their area, length and width parameters were obtained from image analyses using the SmartGrain software [42]. Subsequently, the weight (i.e., mass) was determined and wing loading (i.e., weight-to-area ratio) was also calculated.

To measure the terminal velocity of descent in still air, we used an apparatus based on the original design from Askew et al. [43]. As illustrated in Figure 2, the apparatus consists of a plastic cylinder with two sections and a total height of 270 cm. In the upper preparation section, samaras are released from the top with an unstable transitional rotating motion

(i.e., rapid initial falling), while in the lower measurement section, samaras descend in stable autorotation status (i.e., with terminal velocity). Two optical detectors (i.e., fiber amplifier with an optical resolution of 2 mm and a temporal sensitivity of 100 μs) are respectively set at the entrance and exit of the measurement section to record the time point of passage of the falling samaras. Thus, the duration of period for a samara descending through the measurement section with a stable autorotation motion could be obtained from the time-lag between the two records, and the terminal velocity of samaras was calculated based on the fixed distance of 114 cm.

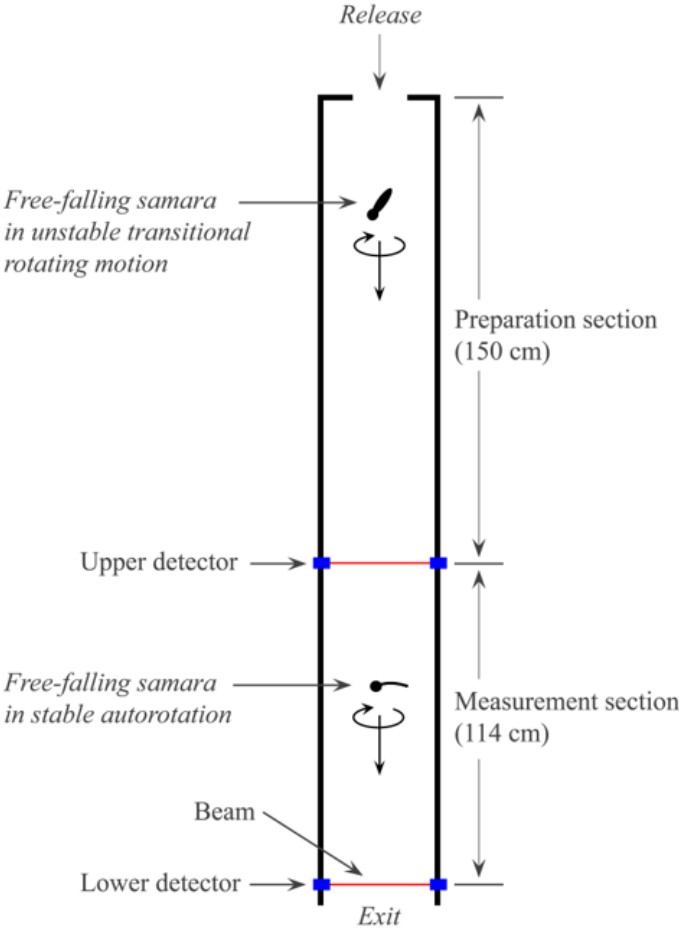

**Figure 2.** Illustration of the apparatus used for the measurement of terminal velocity of samara in the free-falling test.

To minimize the distance of a free-falling samara reaching equilibrium falling (i.e., stable autorotation) from rapid initial falling, all samaras were released with a fixed initial altitude—the downward facing of lower surface—as suggested by Lee and Lee [44]. From prior personal observations, we found that the distance was generally less than 100 cm. Thus, the length of preparation section was set to 150 cm to ensure that samaras can reach to stable autorotation motion before entering the measurement section. In addition, during the test, the whole descending process was monitored to avoid any exceptions (e.g., the occurrence of disequilibrium falling in the measurement section).

### 2.4. Statistical Analyses

The differences in physical/geometric traits (i.e., weight, area, length and width), wing loading, and terminal velocity of samaras collected from *A. palmatum* between open and shaded microhabitats were tested using nested ANOVAs (i.e., linear mixed models with microhabitat light condition as the fixed factor, sampling site and tree replicate as the

random factors). To investigate the impacts of internal (i.e., physical traits) and external (microhabitat light condition) determinants on the dispersal potential of samaras (here represented by terminal velocity), we used nested ANCOVAs (i.e., linear mixed models with physical trait covariate, microhabitat type and their interaction term as fixed factors, and site and tree as random factors). Furthermore, the impacts of microhabitat light condition on the morphological allometry (in terms of area-weight, length-area, width-area, and width-length relationships in log-log scales) of samaras were also investigated based on nested ANCOVAs (i.e., linear mixed models with microhabitat type, allometric covariate and their interaction term as fixed factors, and site and tree as random factors). All the analyses were performed using packages 'lme4' [45], 'lmerTest' [46] and 'LMERConvenienceFunctions' [47] in R v4.1.0 [48].

### 3. Results

Samaras produced by *A. palmatum* trees which grew in shaded microhabitats had a significantly slower terminal velocity and less weight than those produced by trees which grew in open microhabitats (Figure 3A,B). Interestingly, however, microhabitat light condition had no effect on the area, length or width of samaras (Figure 3C–E). This led to a significant lower wing loading of samaras collected from shaded microhabitats (Figure 3F).

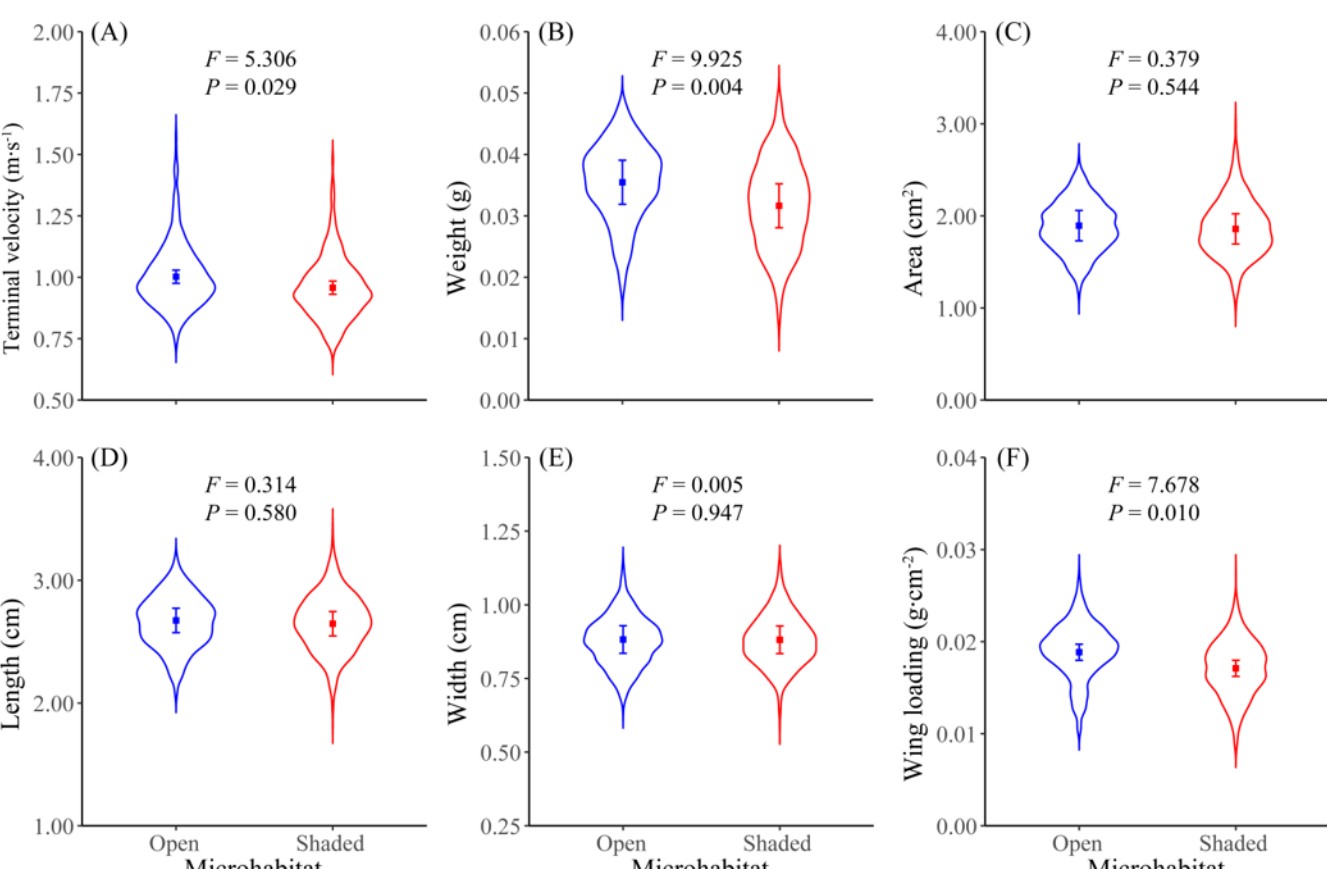

**Figure 3.** Dispersal potential and physical properties of *Acer palmatum* samaras in response to different maternal microhabitat light conditions. The dispersal potential is represented by (**A**) terminal velocity of descent, and the physical properties are (**B**) weight, (**C**) area, (**D**) length, (**E**) width and (**F**) wing loading. The violin plots describe the distributions of original data, and the associated mean (presented as points) and 95% confidential interval of the mean (presented as error bars) are obtained from linear mixed models in which microhabitat type was the fixed factor, and site and tree were the random factors. *F* and *p* values are calculated from *F*-statistics for the linear mixed model using a type III sum of squares with Satterthwaite's method.

Both weight and square root of wing loading (wing-loading$^{0.5}$, hereafter) had significant positive effects on the terminal velocity of samaras; meanwhile the extents of these positive effects were further strengthened in shaded microhabitats as compared to those in open microhabitats (Table 2). In contrast, area, length and width had significant (or marginally significant) negative effects on terminal velocity, and the extents of these negative effects were not significantly affected by microhabitat type (Table 2).

**Table 2.** Summary statistics (estimate, standard error (S.E.), degree of freedom (D.F.), *t*-value, *p*-value) of the nested ANCOVAs (i.e., linear mixed models with microhabitat type, physical property and their interaction term as fixed effects, and site and tree replicate as random effects) on the terminal velocity of samaras.

| Model | Variable | Estimate | S.E. | D.F. | *t* | *p* |
|---|---|---|---|---|---|---|
| 1 | (Intercept) [†] | 0.844 | 0.031 | 1757 | 27.184 | <0.001 |
| | Shaded [‡] | −0.111 | 0.037 | 26 | −2.990 | 0.006 |
| | Weight | 4.473 | 0.721 | 1757 | 6.202 | <0.001 |
| | Shaded × Weight | 2.640 | 0.989 | 1757 | 2.670 | 0.008 |
| 2 | (Intercept) [†] | 1.070 | 0.035 | 1757 | 30.458 | <0.001 |
| | Shaded [‡] | −0.084 | 0.049 | 26 | −1.711 | 0.099 |
| | Area | −0.036 | 0.017 | 1757 | −2.100 | 0.036 |
| | Shaded × Area | 0.021 | 0.024 | 1757 | 0.851 | 0.395 |
| 3 | (Intercept) [†] | 1.106 | 0.062 | 1757 | 17.772 | <0.001 |
| | Shaded [‡] | −0.121 | 0.086 | 26 | −1.403 | 0.173 |
| | Length | −0.039 | 0.023 | 1757 | −1.709 | 0.088 |
| | Shaded × Length | 0.028 | 0.032 | 1757 | 0.899 | 0.369 |
| 4 | (Intercept) [†] | 1.114 | 0.05 | 1757 | 22.360 | <0.001 |
| | Shaded [‡] | −0.060 | 0.072 | 26 | −0.837 | 0.410 |
| | Width | −0.126 | 0.054 | 1757 | −2.329 | 0.020 |
| | Shaded × Width | 0.017 | 0.079 | 1757 | 0.222 | 0.825 |
| 5 | (Intercept) [†] | 0.424 | 0.056 | 1757 | 7.630 | <0.001 |
| | Shaded [‡] | −0.156 | 0.072 | 26 | −2.152 | 0.0410 |
| | Wing-loading$^{0.5}$ | 4.227 | 0.394 | 1757 | 10.732 | <0.001 |
| | Shaded × Wing-loading$^{0.5}$ | 1.067 | 0.534 | 1757 | 1.998 | 0.046 |

In each model, physical property is a numeric variable, while microhabitat type is a categorical variable with [†] open microhabitats as the baseline (or intercept) for the comparison with [‡] shaded microhabitats. *n* = 1789.

The area-weight allometric relationship of samaras were significantly dependent on the light condition of microhabitats. That is, with the same weight, samaras collected from shaded microhabitats tended to have larger areas than those collected from open microhabitats, and the extent of this difference increased with increasing weight (Figure 4A). Interestingly, although the length-area allometry was fully independent of microhabitat light condition (Figure 4B), the width-area and width-length allometries significantly relied on the microhabitat light condition. More specifically, only in the range of small sizes, samaras collected from shaded microhabitats had a greater width than those collected from open microhabitats, given the same area or length (Figure 4C,D).

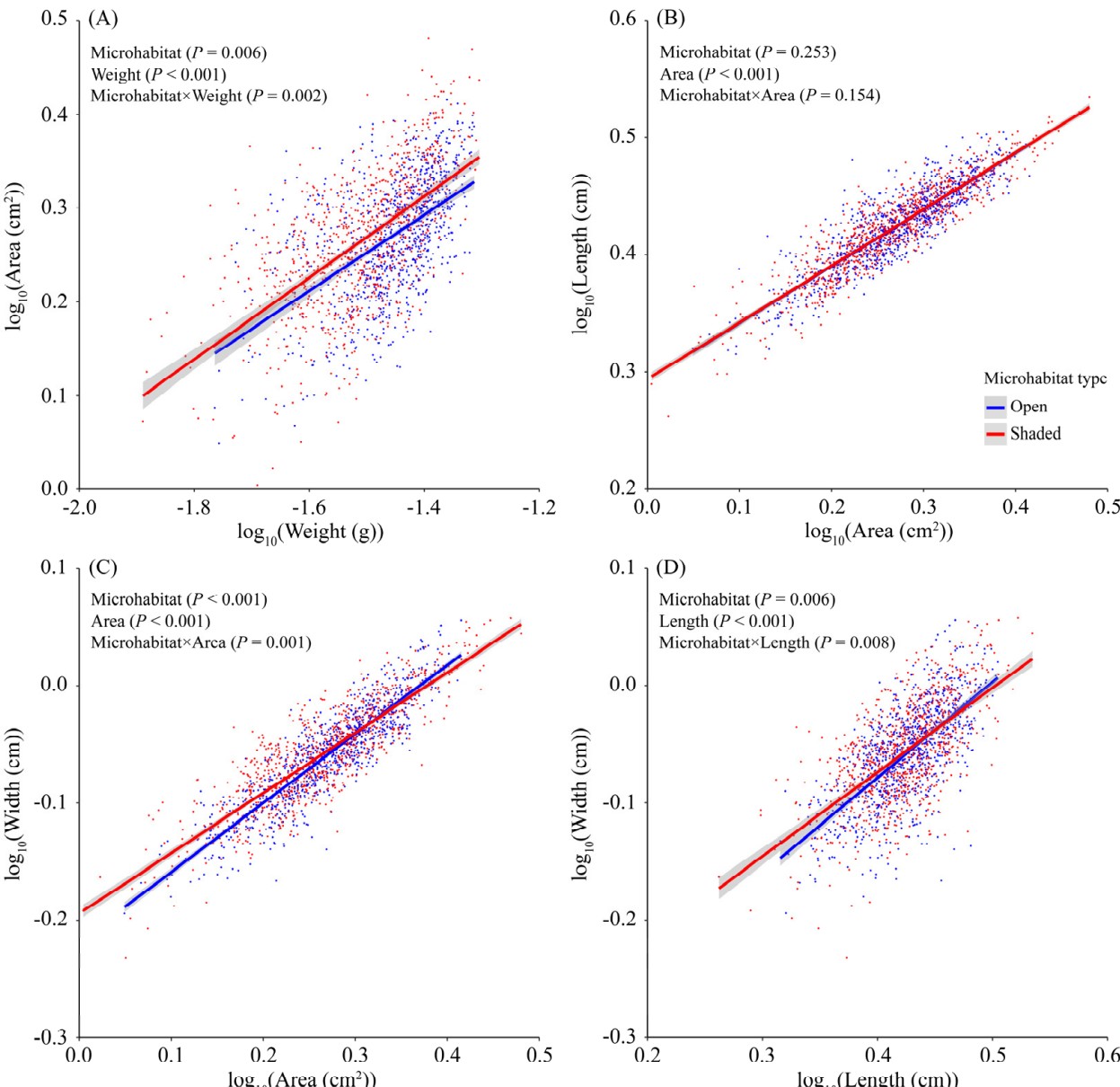

**Figure 4.** Morphological allometries of samaras of *Acer palmatum* trees in response to different maternal microhabitat light conditions. The allometries include (**A**) area-weight, (**B**) length-area, (**C**) width-area, and (**D**) width-length relationships in log-log scales. In each analysis, log10-transformed properties in y-axis and x-axis are respectively set as the dependent variable and the covariate. In the corresponding linear mixed model, microhabitat type, covariate and their interaction term are the fixed factors, and site and tree are the random factors. The significance of the fixed effects is presented here as *p* values, which were calculated based on type-III analysis-of-variance.

## 4. Discussion

By examining the physical/geometric characteristics and terminal velocity of nearly 1800 *A. palmatum* samaras collected from two microhabitat types with contrasting light conditions, the results confirmed our hypothesis that samaras collected from shaded microhabitats indeed had greater dispersal potential than those collected from open microhabitats. Since a heavily shaded microhabitat entails a greater possibility of being shaded by top-canopy trees and probably also higher neighborhood density, thus more intensive competitions from both above- and belowground for the next generation, a greater dispersal potential of samaras produced by maternal trees in shaded microhabitats may help offspring to escape from intensive competition, and to have a better chance to find less competitive microhabitats for growth. Such a dispersal advantage can be largely attributed

to the morphological plasticity of samaras produced by the maternal trees. In general, samaras produced in shaded microhabitats tend to have a lower mean wing-loading by reducing mean weight. However, this reduction in weight was not associated with a reduction in area. In allometric details, on the one hand, in the large size range, samaras collected from shaded microhabitats tended to have larger areas than those collected from open microhabitats; on the other hand, in the small size range, samaras collected from shaded microhabitats tended to have wider wings. Both strategies significantly reduced terminal velocity. Thus, these findings suggest that the improved dispersal potential reflects an active maternal control in response to microhabitat light condition via a change in samara morphological allometry.

As most wind-dispersed plants, the family of maple species are generally less successful in competition for sites, and tend to have a scattered distribution in forests [49]. The improved dispersal potential of samaras produced by *A. palmatum* in shaded microhabitats thus can be well explained by the habitat selection theory [17] under the context of maternal environmental effects. That is, to cope with the sessile lifestyle, plants are evolved to make good use of current cues to predict biotic and abiotic environmental conditions in future [50,51]. The impacts of such anticipatory behaviors have been extensively demonstrated in their own generation, e.g., root over-proliferation for soil resource pre-emption in response to the presence of neighbors' roots before the decline of resource availability [52], and autumn senescence in response to the reduction of photoperiod before the onset of winter [53]. Meanwhile, there is also evidence suggesting that the impacts even can be transmitted to the next generation. For example, in the presence of a competitor which carries a message of higher likelihood of dense seedling stands, maternal plants of *Phaseolus vulgaris* tend to produce seeds with larger sizes that convey competitive advantages in offspring generation [54]. Similarly, in our case, a shaded microhabitat experienced by the maternal trees implies a greater chance of being heavily shaded at the home site or nearby for offspring. Thus, in response to shaded microhabitats, these maternal trees can have two alternative adaptive strategies: (i) producing heavier samaras with lower dispersal potential but which convey competitive advantages to seedlings in home sites, or (ii) producing lighter samaras with higher dispersal potential which enable these samaras to be dispersed further away for a higher probability of finding unshaded sites. Obviously, our studied *A. palmatum* trees had adopted the second strategy.

So far, maternal environmental effects on the morphology and/or dispersal potential of seeds have been mainly tested on non-samara species and mostly in controlled or at least manipulated experiments [12]. For instance, Larios and Venable [55] found that seeds of ballistically-dispersed species *Dithyrea californica* had smaller sizes and longer dispersal distances when the maternal plants were grown in crowed stands. Weiner [50] showed that maternal plants of ballistic species *Centaurea maculosa* grown in infertile soil produced longer fruits than those grown in fertile soil, and the length of fruits was positively correlated with dispersal distance of seeds. Thus, even when there was no difference in seed size, seeds from infertile soil still had larger dispersal distances. Furthermore, Martorell and Martínez-López [20] showed that the proportional production of awned achenes (animal-dispersed) increased and that of unawned achenes (non-dispersible) decreased with the increase of neighbor density and water stress experienced by maternal plants of the heteromorphic species *Heterosperma pinnatum*. Aforementioned examples all echo our field investigation-based findings of maternal *A. palmatum* trees actively assisting offspring in escaping from unfavorable home sites.

It is interesting to note that a study of an old field succession from Peroni [37] showed that samaras produced by *A. rubrum* from early successional populations had heavier weights, larger wings, and importantly, also a lower wing-loading than those produced by trees from late successional populations. Since the extent of light interception from forest top-canopy layer should be positively correlated with the successional stage, the finding of a lower wing-loading (indicating greater dispersal potential) in early successional populations seemed to be contradictive to ours. This lower wing-loading might simply

be due to the fact that early successional *A. rubrum* trees have more available resources to produce larger samaras, and an intrinsic weight-area allometric relationship determines a faster growth rate in area than in weight along the development of samaras. Alternatively, it might be that during succession, natural selection only favors colonizing traits (i.e., more dispersible) in newly established but not in mature populations [37]. Unlike the maternal effects that are based on non-Mendelian mechanisms, natural selection is a process based on genetic inheritance (i.e., Mendelian mechanisms) [18,38]. However, it is still unclear why the two ecological forces drove the adaptative strategies of plants into opposite directions in response to the same or at least quite similar environmental conditions.

It is also worth noting that since *A. palmatum* is a shade tolerant species, it might even be possible that full sunlight conditions are more unfavorable/stressful than shaded conditions for this species. However, to what extent this possibility can hold true in our case is still questionable. This is because our findings of lower weight and higher dispersal potential in samaras produced by trees in shaded microhabitats, as compared to those in open microhabitats, are very consistent with the escape strategy in response to unfavorable home sites predicted by habitat selection theory and informed dispersal theory. Moreover, during the investigation, we found trees in shaded microhabitats demonstrating typical signs of resource shortage, i.e., less branches, lower leaf densities, and also less samara production than trees in open microhabitats.

In general, traits (except for size or weight) of seeds, especially those related to dispersal, have received much less attention as compared to vegetative traits in plant science [10,15]. In the limited research on intraspecific morphological variation of winged seeds in response to environmental variations (from either maternal effect or natural selection perspective), studies generally measure no more than weight, area and wing-loading parameters (e.g., [14,37,56]). However, results from aerodynamic studies indicate that the variation of wing-loading$^{0.5}$ can only partially explain the variation of terminal velocity of samaras [23,36], and the shape (e.g., length, width and their ratio), 3D features and texture details are also critical determinants [57]. Yasuda and Azuma [33] even demonstrated that the curvature and surface roughness of wings are more influential than wing-loading on the descending performance of samaras. Therefore, these dispersal-related 3D geometric features beyond the weight and area of samaras may be also under the selection of the maternal environmental effects, and deserve more attention in future research.

Finally, it should be noted that a slower terminal velocity does not necessarily always result in a longer dispersal distance, especially in nature [58–60]. Mounting evidence from field-based investigations (or seed release experiments) suggests that the height of maternal plants (or more specific, seed release height) is more deterministic than terminal velocity on the dispersal distance of windborne seeds [61,62], including samaras [59]. For instance, Augspurger et al. [63] showed that a good model for predicting dispersal of windborne seeds in field conditions must include maximal tree height but does not need wing-loading$^{0.5}$, which only explained less than 5% of the variation in dispersal distance. Teller et al. [21] found that a greater dispersal potential of *Carduus nutans* seeds in response to drought stress in the maternal site was not caused by any change in seed morphology but due to a taller status of the maternal plants. In addition, environmental factors are also critical for wind dispersal in fields [61]. This, at least, includes wind condition [64] and surrounding vegetation structures (e.g., the density and distribution of boles, branches and leaves), which not only influence local wind conditions [59] but also generate collision events [65]. These findings suggest that in the study of how maternal environmental effects operate on seed dispersal performance should not only focus on seed characteristics per se, but also need to take other maternal traits (e.g., height and crown shape) and environmental factors into account.

## 5. Conclusions

By conducting a field-based investigation on the descending performance and associated morphological traits of circa 1800 samaras collected from 30 trees of *A. palmatum*

growing in two contrasting microhabitat light conditions, we showed that samaras from shaded microhabitats had a slower terminal velocity of descent, thus a greater dispersal potential, than those from open microhabitats. Such a dispersal advantage can be largely attributed to maternally-controlled changes in the morphological allometries of samaras. As predicted by habitat selection theory and informed dispersal theory, our findings suggested that the dispersal performance of samaras were strongly influenced by their home site conditions. Maternal trees were actively involved in this process, as an adaptation to increase the probability of dispersing offspring to suitable sites for better growth and performance.

**Author Contributions:** Conceptualization, B.J.W.C.; methodology, B.J.W.C.; software, B.J.W.C., X.W. and Y.D.; validation, X.W. and Y.D.; formal analysis, B.J.W.C.; investigation, X.W. and Y.D.; data curation, B.J.W.C., X.W. and Y.D.; writing—original draft preparation, B.J.W.C.; writing—review and editing, H.J.D. and N.P.R.A.; visualization, B.J.W.C.; supervision, B.J.W.C. and X.X.; project administration, B.J.W.C. and X.X.; funding acquisition, B.J.W.C. All authors have read and agreed to the published version of the manuscript.

**Funding:** This research was sponsored by National Natural Science Foundation of China (32071526), and Qing Lan Project of Jiangsu Province of China.

**Institutional Review Board Statement:** Not applicable.

**Informed Consent Statement:** Not applicable.

**Data Availability Statement:** The data presented in this study are available in the article. Additional data are available on request from the corresponding author.

**Acknowledgments:** We thank Bin Zhan, Yang Tao and Caiyun Fu for their assistance in data collection.

**Conflicts of Interest:** The authors declare no conflict of interest.

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
