# Peer review of "Maternal Environmental Light Conditions Affect the Morphological Allometry and Dispersal Potential of Acer palmatum Samaras"

_forests, doi:10.3390/f12101313_

Round 1
Reviewer 1 Report
This is a very interesting study examining a very interesting question that is rarely assessed in the literature. Maternal environmental effects are an important but often neglected component of the seed dispersal process in wild conditions. The manuscript is clear and well-written and the science behind is sound. I only have a few minor comments and edits below.
(1) Comments
- I can’t see the significant differences in the plots of Figures 3 and 4
- Please provide the geographic coordinates of the sampling sites, it is very useful.
- Please name and cite all R packages used, not just the R program.
(2) Specific details
Lines 43-44: Move ‘community’ before ‘structure’ and delete ‘of’
Line 63: Replace ‘sedentary’ by ‘sessile’
Lines 205-206: Correlation results are missing (i.e., r and P values)
Author Response
Point 1: This is a very interesting study examining a very interesting question that is rarely assessed in the literature. Maternal environmental effects are an important but often neglected component of the seed dispersal process in wild conditions. The manuscript is clear and well-written and the science behind is sound. I only have a few minor comments and edits below.

Response 1: Great thanks for your appreciation!
Point 2: I can’t see the significant differences in the plots of Figures 3 and 4
Response 2: We believe that the concerns raised here are mainly due to the fact that the distribution of data points in Figures 3 and 4 are caused by the effects not only from the microhabitat type (i.e. the focal factor) but also from the sites and trees selected (i.e. other extra unwanted factors). In order to better unveiling the effect of focal factors independently of the influence from sites and trees, we analysed the data using linear mixed models with focal factors as the fixed factors and site and tree as random factors. This has been specifically addressed in the caption of Figures 3 and 4. The results obtained from the linear mixed models are shown as P values of focal fixed factors in Figures 3 and 4.
Point 3: Please provide the geographic coordinates of the sampling sites, it is very useful
Response 3: The geographic coordinates of three sampling sites are given in lines 130-132 (track-change mode). In addition, as suggested by Reviewer 3, the locations where research materials were collected have been pointed on the map (Figure 1). Due to extremely large differences between the scale of microhabitats and the scale of mapped region in Figure 1, it is impossible to mark the detailed boundaries of microhabitats on the map. However, we used different colours of points to indicate the conditions where the materials were collected, see details in Figure 1.
Point 4: Please name and cite all R packages used, not just the R program.
Response 4: The details and also citations of all R packages are provided, see lines 210-211 (track-change mode).
Point 5: Lines 43-44: Move ‘community’ before ‘structure’ and delete ‘of’
Response 5: It is corrected as suggested, see line 44-45 (track-change mode).
Point 6: Line 63: Replace ‘sedentary’ by ‘sessile’
Response 6: It is corrected as suggested, see lines 64 and 285 (track-change mode).
Point 7: Lines 205-206: Correlation results are missing (i.e., r and P values)
Response 7: The correlation interpretation is based on the results of nested ANCOVAs as shown in Table 2. In order to make a clearer and more accurate interpretation, we have rephrased the corresponding sentences, see lines 227-234 (track-change mode).

Reviewer 2 Report
Authors present interesting results in an interesting topic and hey have obtained good data from many samaras distributed among 30 trees. Nevertheless, I think that there are several conceptual and methodological problems.
First of all, it is not possible to argue that differences in samara morphology was due to light conditions. Light is not the only resource that plants need, and microhabitats selected by authors differed in several variables rather than light, since in one environment there is more tree density and then a higher interspecific competition. Therefore, differences in morphology and dispersal potential could not be directly related to light conditions.
Another point is that Acer palmatum is a shade tolerant species within its native range. Therefore, it might be possible that full sun conditions were more stressful than shade conditions. In addition, there are many different Japanese maple cultivars with different shade tolerance. From the text and the context of the study area I suppose that studied maples are planted rather than spontaneous. Authors should confirm this point and indicate the variety or varieties used. At this point, it will be useful to know if trees have gardening cares (i.e., watering, pruning).
Finally, there is a need to know if trees are independent replicates as reflected in the statistical analyses. In other words, within each sampling sites, trees should not be spatially clumped due to their treatment group. More specifically, shaded trees and those in open microhabitat should be interspersed within each sample site (Hurlbert, 1984). This is relevant since it affects the degrees of freedom of the analyses and, therefore, could affect the validity of the results. If trees were clumped in two different zones within each sample site, then a block design should be the right one to analyze the data.
More specific comments.
In fig 3. It is necessary to indicate which statistical analyses were performed to obtain the p-values. Were used sampling site and individuals as random variables? Please, indicate the value of the statistical test (F, t-student).
Table 2 and figure 4. Unify the use of mass and weight.
L237-241. There is no reason to say stressed, see comments above. That trees in different microhabitats produce samaras with different morphology and dispersal potential is a sound result. Later you may discuss the potential reason and consequences. In addition, microhabitats should be described in a neutral way. Open and shaded microhabitat is ok, but I think there is no reason to say that light or shade is the variable that produce the response of the plant.
L262-268. Light is not the only resource plants compete for (see L259-261). In addition, authors used an example of increase in seed weight with competition to justify that studied species decrease seed weight with competition. Perhaps, to produce larger samaras with lower dispersal distance but which convey competitive advantages to the seedlings could be also a viable strategy.
Hurlbert, S. H. (1984). Pseudoreplication and the design of ecological field experiments. Ecological monographs, 54(2), 187-211.
Author Response
Point 1: Authors present interesting results in an interesting topic and hey have obtained good data from many samaras distributed among 30 trees. Nevertheless, I think that there are several conceptual and methodological problems.

Response 1: Great thanks for the appreciation and also suggestions, which have significantly improved our manuscript.
Point 2: First of all, it is not possible to argue that differences in samara morphology was due to light conditions. Light is not the only resource that plants need, and microhabitats selected by authors differed in several variables rather than light, since in one environment there is more tree density and then a higher interspecific competition. Therefore, differences in morphology and dispersal potential could not be directly related to light conditions.
Response 2: We fully agree with the reviewer that a shaded environment also implies a higher probability of more tree density which will generate intensive competitions for resources from both above- and belowground. Therefore, we add this important viewpoint in both Introduction and Discussion sections, see lines 97-103 and 264-270 (track change mode).
Point 3: Another point is that Acer palmatum is a shade tolerant species within its native range. Therefore, it might be possible that full sun conditions were more stressful than shade conditions. In addition, there are many different Japanese maple cultivars with different shade tolerance. From the text and the context of the study area I suppose that studied maples are planted rather than spontaneous. Authors should confirm this point and indicate the variety or varieties used. At this point, it will be useful to know if trees have gardening cares (i.e., watering, pruning)
Response 3: We agree that since A. palmatum is a shade tolerant species, it might be even possible that full sun conditions are more stressful than shaded conditions for this species. However, to what extent this possibility can hold true in our case is still questionable. This is because our findings of lower weight and higher dispersal potential in samaras produced by trees in shaded microhabitats, as compared to those from open microhabitats, are very consistent with the escape strategy in response to unfavourable home sites predicted by habitat selection theory and informed dispersal theory. Moreover, during the investigation, we did find trees in shaded microhabitats demonstrated typical signs of resource shortage, i.e. less branches, lower leaf densities and also less samara production, as compared to trees in open microhabitats. However, we are pleased to add this in the Discussion section, see lines 335-344 (track-change mode).
We also agree that there are many different Japanese maple cultivars with different shade tolerance. In fact, even trees from the same cultivar may still be different in shade tolerance. Of course, trees among different sampling sites (a university campus, a green belt of a road, and a public park) can be different cultivars, but we believe that there is a much higher probability that trees within a site should be bought and planted in the same batch, thus being the same cultivar. Even being the same cultivar, trees within a site should still be different in their genetic backgrounds. Therefore, we took both the site and tree (selected tree individuals) as the random factors in the statistical models (see the structure details in lines 197-211 in track-change mode). And the difference in shade tolerance between cultivars and between different genotypes within a cultivar can be well considered in our analyses.
It is indeed that studied maples are planted rather than spontaneous. We confirmed this point in lines 146-147 (track-change mode). It is unfortunately that we were unable to identify the cultivar type of all maple trees in the three sampling sites, and we were unable to find the relevant information from the internet or administrations either. As planted trees in the urban areas, they do receive gardening cares, of course, the frequency and intensity should be different among sites. We also confirmed this in lines 147-149 (track-change mode).
Point 4: Finally, there is a need to know if trees are independent replicates as reflected in the statistical analyses. In other words, within each sampling sites, trees should not be spatially clumped due to their treatment group. More specifically, shaded trees and those in open microhabitat should be interspersed within each sample site (Hurlbert, 1984). This is relevant since it affects the degrees of freedom of the analyses and, therefore, could affect the validity of the results. If trees were clumped in two different zones within each sample site, then a block design should be the right one to analyze the data.
Response 4: Thanks for the suggested paper from Hurlbert (1984), it has been cited in line 144-146 (track-change mode). In all statistical analyses (using linear mixed models), tree individual together with site (including a university campus, a road, and a public park) were treated as random factors. This is because we believe the genetic difference among individual trees may also affect the weight and shape, thus descending behaviour of samaras. And indeed, by comparing two models that one with tree individual as a random factor and one without tree random factor, we found that the model containing tree random factor worked significantly better. For example, when we analysed the area-weight allometric relationship of samaras in response to different microhabitat types using R:
> mod0 <- lmer(lgarea ~ lgweight*microhabitat + (1|site/tree), data = samara.data)
> mod1 <- lmer(lgarea ~ lgweight*microhabitat + (1|site), data = samara.data)
> anova(mod0, mod1)
Data: samara.data
Models:
mod1: lgarea ~ lgweight*microhabitat + (1|site)
mod0: lgarea ~ lgweight*microhabitat + (1|site/tree)
npar AIC BIC logLik deviance Chisq Df Pr(>Chisq)
mod1 6 -5420.2 -5387.2 2716.1 -5432.2
mod0 7 -6021.9 -5983.5 3018.0 -6035.9 603.76 1 < 2.2e-16 ***
Point 5: In fig 3. It is necessary to indicate which statistical analyses were performed to obtain the p-values. Were used sampling site and individuals as random variables? Please, indicate the value of the statistical test (F, t-student).
Response 5: Indeed, sampling site and tree individuals were used as random variables in the analyses. The details of statistical methods have been provided in the caption of Figure 3, see lines 223-226 (track-change mode). The F and P values of the statistical tests are also indicated in the figure.
Point 6: Table 2 and figure 4. Unify the use of mass and weight.
Response 6: They are unified as weight, see corrections in Table 2.
Point 7: L237-241. There is no reason to say stressed, see comments above. That trees in different microhabitats produce samaras with different morphology and dispersal potential is a sound result. Later you may discuss the potential reason and consequences. In addition, microhabitats should be described in a neutral way. Open and shaded microhabitat is ok, but I think there is no reason to say that light or shade is the variable that produce the response of the plant.
Response 7: As suggested, a brief discussion on the potential reason and consequences has been added in lines 264-270 (track-change mode). We understand the concerns from the reviewer that there is no reason to say that light or shade is the variable that produce the response of the plant. So, in the revised manuscript, we described microhabitats in a more neutral way.
Point 8: L262-268. Light is not the only resource plants compete for (see L259-261). In addition, authors used an example of increase in seed weight with competition to justify that studied species decrease seed weight with competition. Perhaps, to produce larger samaras with lower dispersal distance but which convey competitive advantages to the seedlings could be also a viable strategy.
Response 8: We agree that light is not the only resource plants compete for, this has been admitted in lines 97-103 and 264-270 (track change mode). We are pleased to add the suggestion that “to produce larger samaras with lower dispersal distance but which convey competitive advantages to the seedlings could be also a viable strategy” in lines 297-303 (track change mode) to further improve the discussion.
Point 9: Hurlbert, S. H. (1984). Pseudoreplication and the design of ecological field experiments. Ecological monographs, 54(2), 187-211.
Response 9: Thanks for providing the paper, it has been read and also cited, see line 145 (track change mode).

Reviewer 3 Report
The review of the submitted manuscript entitled Maternal environmental light conditions affect the morphological allometry and dispersal potential of Acer palmatum samaras.
The submitted manuscript describes the research results into the differences in the dispersal potential of Acer palmatum samaras. This study was carried out on fruits collected from open and shaded habitats. It has been shown that the seeds produced in a shaded habitat have a greater dispersion capacity than those harvested from plants growing along the road. The manuscript is perfectly prepared. The Introduction describes the research problem well. The goals and study hypotheses are rational and well-motivated by the knowledge of the research conducted so far. The materials and methods use simple statistics, the results of which are described understandably. Nice Discussion!
If the authors have such a possibility, they should better document the sites where the research material is collected, i.e. points on the maps. Research areas marked as shady habitats on the maps present themselves as areas with a predominance of semi-open habitats.
L.29-30: The keywords contain the words used in the title. I suggest that you replace them with other ones that will inform about the article's content.
Kind regards
Author Response
Point 1: The submitted manuscript describes the research results into the differences in the dispersal potential of Acer palmatum samaras. This study was carried out on fruits collected from open and shaded habitats. It has been shown that the seeds produced in a shaded habitat have a greater dispersion capacity than those harvested from plants growing along the road. The manuscript is perfectly prepared. The Introduction describes the research problem well. The goals and study hypotheses are rational and well-motivated by the knowledge of the research conducted so far. The materials and methods use simple statistics, the results of which are described understandably. Nice Discussion!

Response 1: Great thanks for your appreciation!
Point 2: If the authors have such a possibility, they should better document the sites where the research material is collected, i.e. points on the maps. Research areas marked as shady habitats on the maps present themselves as areas with a predominance of semi-open habitats.
Response 2: The locations (the position of selected maple trees) where research materials were collected have been pointed on the map (see Figure 1). Due to extremely large differences between the scale of microhabitats and the scale of mapped region in Figure 1, it is impossible to mark the detailed boundaries of shady microhabitats on the map. However, we used different colours of points to indicate the conditions where the materials were collected, see the caption of Figure 1.
Point 3: L.29-30: The keywords contain the words used in the title. I suggest that you replace them with other ones that will inform about the article's content.
Response 3: The keywords used in the title have been replaced by other ones, such as “autorotation”, “Japanese maple”.

Round 2
Reviewer 2 Report
Authors have completed all the changes suggested by the review.